# Association of Differences in Dietary Behaviours and Lifestyle with Self-Reported Weight Gain during the COVID-19 Lockdown in a University Community from Chile: A Cross-Sectional Study

**DOI:** 10.3390/nu13093213

**Published:** 2021-09-16

**Authors:** Addi Rhode Navarro-Cruz, Ashuin Kammar-García, Javier Mancilla-Galindo, Gladys Quezada-Figueroa, Mariana Tlalpa-Prisco, Obdulia Vera-López, Patricia Aguilar-Alonso, Martín Lazcano-Hernández, Orietta Segura-Badilla

**Affiliations:** 1Departamento Bioquímica-Alimentos, Facultad de Ciencias Químicas, Benemérita Universidad Autónoma de Puebla, Puebla 72570, Mexico; addi.navarro@correo.buap.mx (A.R.N.-C.); obdulia.vera@correo.buap.mx (O.V.-L.); patricia.aguilar@correo.buap.mx (P.A.-A.); lazmar@gmail.com (M.L.-H.); 2Sección de Estudios de Posgrado e Investigación, Escuela Superior de Medicina, Instituto Politécnico Nacional, Mexico City 11340, Mexico; kammar_nutrition@hotmail.com (A.K.-G.); marianatp967@gmail.com (M.T.-P.); 3Departamento de Atención Institucional Continua y Urgencias, Instituto Nacional de Ciencias Médicas y Nutrición Salvador Zubirán, Mexico City 14080, Mexico; 4Unidad de Investigación UNAM-INC, Instituto Nacional de Cardiología Ignacio Chávez, Mexico City 14080, Mexico; javimangal@gmail.com; 5Facultad de Medicina, Universidad Nacional Autónoma de México, Mexico City 04360, Mexico; 6Programa UBB Saludable, Departamento de Nutrición y Salud Publica, Facultad de Ciencias de la Salud y de los Alimentos, Universidad del Bío-Bío, Avda. Andrés Bello 720, Chillán 3780000, Chile; glquezada@ubiobio.cl

**Keywords:** COVID-19, feeding behaviour, home confinement, lifestyles, weight gain

## Abstract

Domiciliary confinement of people is one of the main strategies to limit the impact of COVID-19. Lockdowns have led to changes in lifestyle, emotional health, and eating habits. We aimed to evaluate the association of differences in dietary behaviours and lifestyle with self-reported weight gain during the COVID-19 lockdown in Chile. In this cross-sectional analytical study, five previously validated surveys were condensed into a single 86-item online questionnaire. The survey was sent to 1000 potential participants of the university community; it was kept online for 28 days to be answered. Of the 639 respondents, the mean self-reported weight gain during confinement was 1.99 kg (standard deviation [SE]: 0.17) and 0.7 (SE: 0.06) units of body mass index (BMI) (both *p* < 0.001) and the median difference in body weight during lockdown was 3.3% (interquartile range [IQR]: 0.0–6.7). The differences of intake of most food groups before and during lockdown were associated with greater self-reported weight, BMI and percentage weight gain. Differences in lifestyle (odds ratio [OR] = 14.21, 95% confidence interval [95%CI]: 2.35–85.82) worsening eating habits (OR = 3.43, 95%CI: 2.31–5.09), and more consumption of sweet or filled cookies and cakes during lockdown (OR = 2.11, 95%CI: 1.42–3.13) were associated with self-reported weight gain. In conclusion, different dietary behaviours (mainly consumption of industrialized foods) during lockdown, as well as quality of life deterioration were the main factors associated with self-reported weight gain during lockdown.

## 1. Introduction

One important aspect of the study of epidemics is understanding how societies react to contagious diseases [1]. Lockdowns at the country and regional levels have been one of the first strategies to limit the spread of the severe acute respiratory syndrome coronavirus 2 (SARS-CoV-2) and coronavirus disease (COVID-19) [2]. It is estimated that nearly 4 billion people have self-quarantined at home during the pandemic, which could result in a high prevalence of psychological distress, manifested as moodiness and irritability, emotional disturbances, disturbed sleep and diet, post-traumatic stress, and depressive symptoms [3,4,5]. The impact of these restrictions on health behaviours and lifestyles remains to be fully characterised, although some studies have started to address this globally [6,7,8,9,10,11].

In a cohort of university students from Eastern Asia changes in body mass index after lockdown occurred in 54% (32% increased body mass index (BMI) and 22% lost weight) [12]. Worsening mental health, sleep, dietary, and other lifestyle components markedly occurred in those who experienced weight gain. In a Chinese cohort of university students, an overall increase in weight during the 4-month lockdown occurred [13]. Secondary school students from Poland reported better dietary habits during lockdown, although changes in weight could not be assessed in this study [14]. Interestingly, even when students and academics affiliated to Nutrition Societies in Spain reported better eating habits during the COVID-19 lockdown, an overall increase in self-reported body weight was noted [15].

Altogether, these studies show that COVID-19 lockdowns in different countries have impacted lifestyles and changes in weight differently and that changes in body weight status are apparently determined by multiple components of lifestyle and eating behaviours which may vary across continents due to sociocultural, economic, and idiosyncratic differences. However, few studies have attempted to holistically address the impact of such components on weight gain during lockdowns.

Studying which factors are associated with short-term weight gain during lockdowns would be important since short-term overfeeding and weight gain are known to have a negative impact on health [16]. Further understanding of how lockdowns are associated with different lifestyle, feeding, and emotional behaviours as well as factors associated with weight gain during lockdowns could be important to conceive strategies to limit their impact on subsequent lockdowns due to COVID-19 or other disease outbreaks that may require domiciliary confinement of people.

In this study, we sought to evaluate the association of differences in dietary behaviours and lifestyles on self-reported weight gain of students, academics, and administrative members of the University of Bío-Bío during the COVID-19 pandemic-derived lockdown in Chile.

## 2. Materials and Methods

### Study Design and Participants

We conducted a cross-sectional analytical study consisting of an online survey delivered to students, administrative officials and teachers of both sexes and all ages of the University of Bío-Bío, Chile. Five surveys were condensed into a single 86-item questionnaire that included: (1) general and sociodemographic history, (2) self-reported eating habits before and during domiciliary confinement [17], (3) measurement of emotional feeding behaviour [18], (4) lifestyle before and during confinement [19], and (5) food insecurity [20]. The survey was delivered via Google Forms and was priorly validated by experts in nutrition and public health using a sample of 57 participants.

Users of the institutional electronic mailing system from the University Community of Bío-Bío were eligible to be included in this study. People who were not part of the educational community or who did not give their consent to participate in the study were excluded. The study was developed within the framework of the DIUBB 191220 3/R project, with the approval of the Bioethics and Biosafety Committee of the University of Bío-Bío. Once informed consent was obtained, participants were able to access the survey that was kept online for 28 days (from 21 July to 19 August 2020). One thousand participants were invited to complete the survey. The complete survey is provided in the Appendix A.

The main exposures assessed in this study were self-reported eating habits before and during lockdown, emotional feeding behaviour, lifestyle before and during lockdown, and food safety. These exposures and their association with self-reported weight gain, differences in BMI, and percentage of self-reported weight gain were evaluated. Different questionnaires and evaluations were used to assess these exposures.

The general and sociodemographic history collected in the survey were the following: place of residence, type of household, age, occupation, sex, level of studies, and university campus of origin. Participants were asked to place within the questionnaire their usual weight as measured before confinement, and they were asked to weigh themselves within a couple of days before answering the questionnaire, both weights were reported as independent values. Weight measurement was requested to be carried out with prior 8-h fasting and with the support of a companion who would take weight measurements while the participant maintained an erect position and frontal gaze.

For the measurement and evaluation of eating habits before and during lockdown, a Food Consumption Frequency Questionnaire (CFCA, for its acronym in Spanish) [17] was applied in which different groups of foods typically consumed in the Chilean population were included. Participants were given two CFCA to answer: the first corresponded to pre-lockdown food consumption, and the second questionnaire, during-lockdown food consumption.

For the measurement of emotional feeding behaviour, defined as eating in response to a range of negative emotions, such as anxiety, depression, anger, and loneliness, we used the Emotional Eating Survey [18] adapted to the Chilean population [21]. Responders were classified into: no emotional feeding behaviour (0 to 5 points), little emotional feeding behaviour (6 to 10 points), some emotional feeding behaviour (11 to 20 points), and emotional feeding behaviour (21 to 30 points).

To evaluate lifestyle before and during lockdown, we used the Fantastic questionnaire (FQ) [19], comprising 10 sections: family and friends, relationships and physical activity, nutrition, tobacco, alcohol, and other drug consumption, sleep and stress, work and personality type, introspection, control of health and sexual behaviour, and other behaviours. The following scores were used: 0 to 46 points corresponded to the danger zone; 47 to 72, could be better; 73 to 84, adequate; 85 to 102, right path; and 103 to 120, fantastic lifestyle.

We used the Household Food Insecurity Access Component Scale (HFIAS) [20] to classify food safety into 4 categories: 1 = safe, 2 = mildly unsafe, 3 = moderately unsafe, and 4 = severely unsafe.

Self-reported pre-lockdown and during-lockdown weights were collected, and self-reported weight gain or weight loss were defined according to the difference between both measurements. Similarly, pre-lockdown and during-lockdown differences in BMI were calculated based on self-reported weight and height. Percentage weight gain was calculated by dividing self-reported weight gain between self-reported pre-lockdown weight multiplied by 100.

Differences in dietary intake were defined according to self-reported frequency of consumption of foods during-lockdown, compared with that before lockdown, based on an ordinal scale including three possible options: lower, same, and higher. Differences in lifestyle were based on self-reported pre-lockdown and during-lockdown responses and classified as positive when a higher during-lockdown score in the FQ led to a higher lifestyle category compared to the pre-lockdown category; negative, when the score led to a lower during-lockdown category, and when the category did not change. Worsening eating habits were defined when self-reported eating habits were worse during-lockdown compared to pre-lockdown.

## 3. Statistical Analysis

Descriptive data are shown as frequencies and percentages, and as mean with standard deviation (SD) or standard error (SE), or median with 25th–75th percentiles. Normal distribution was verified for every variable through kurtosis (±2), asymmetry (±0.5), and the Kolmogorov–Smirnov test was applied. Therefore, parametrical statistics were applied to compare quantitative variables. Quantitative comparisons were made with *t*-test, ANOVA one-way with Dunnett post hoc test, and qualitative comparisons were performed through X^2^, X^2^ for trend or McNemar test.

Different linear regression models were applied to determine the ability of a higher self-reported food intake to predict self-reported weight, BMI, and percentage weight differences between during-lockdown and pre-lockdown. The Stepwise Forward method was applied for the introduction of variables with a *p* < 0.1 in every step of the model. Variables with a *p* < 0.05 were included in the final linear regression model. The final models were adjusted by age (continuous variable), sex, and occupation.

A multivariable logistic regression model was created to estimate the odds of experiencing a difference in self-reported during-lockdown body weight of ≥5%. The first multivariable model was performed with the Stepwise Forward method. The final model was created by the Enter method in which the variables that had a statistical significance of *p* < 0.1 in the prior model were included. The estimation of the effect size of every variable in the model was adjusted for sex, age (continuous variable), and occupation. The model was validated by Akaike and Bayesian information criteria, Hosmer–Lemeshow statistic, and the area under the curve (AUC). A multinomial logistic regression model was created to determine the odds of experiencing a positive difference of 5–9% or ≥10% in self-reported during-lockdown weight associated with more self-reported food intake during lockdown. The variables included in this model were determined by the Enter method and the results were adjusted for sex, age (continuous variable), and occupation. For all linear regression models, data are presented as regression coefficients, whereas the results of logistic regression models are provided as odds ratio (OR) with their respective 95% confidence intervals (95%CI). The assumptions of the regression analyses were verified by the residual analysis and collinearity analyses.

We created a random effects model in which we determined the differences in self-reported weight, BMI and percentage weight gain during lockdown associated with differences in lifestyle and emotional feeding behaviour during lockdown. The effect size of each factor and that of the interaction was calculated with partial eta squared. We plotted the means and their 95%CI for each level within lifestyle differences. A post hoc test of Bonferroni was applied to make pairwise comparisons between levels of emotional feeding behaviour.

No missing data were found in the sample. A value of *p* < 0.05 was considered as statistical significance. All statistical analyses were performed in the SPSS statistical software v.21. Graphs were created in GraphPad Prism v.9.1.1.

## 4. Results

Out of 1000 potential survey participants, 639 answered the questionnaire; 66.8% were women. The mean age of participants was 28.9 (SD: 13.2) years, with an age range of 18 to 88 years. Descriptive characteristics of the participants are provided disaggregated for sex (Table 1), and according to self-reported percentage weight gain (Appendix A). The mean self-reported weight gain during confinement was 1.99 (SE: 0.17) Kg and 0.7 (SE: 0.06) units of BMI (both *p* < 0.001) (Appendix A), and the median of difference in self-reported body weight was 3.3% (IQR: 0.0–6.7). Out of 351 participants who self-reported a normal weight before lockdown (IMC < 24.9), 19.9% (*n* = 70) self-reported overweight (IMC 25–29.9) during-lockdown, and none reported obesity (IMC ≥ 30). Out of 182 who reported being overweight before lockdown, 17.6% (*n* = 32) reported metrics classifying them as having obesity during-lockdown (*p* < 0.001). In Appendix A we show the frequencies of dietary consumption for every food group before and during lockdown. More self-reported dietary intake during lockdown of most food groups was associated with a greater self-reported during-lockdown weight, BMI and percentage weight gain. The food groups with the highest content of simple carbohydrates, refined sugars, and saturated fats had the greatest differences observed (Table 2).

We created 3 different prediction models for greater self-reported weight, BMI, and percentage weight gain during lockdown. In all models, processed foods and foods rich in saturated fats and sugars (sweet or stuffed cookies, cakes, salty snacks, processed meats and sausages, white or brown sugar, rice, potatoes, noodles or quinoa, sugary drinks or juices) had the best predictions of self-reported weight (R^2^ = 0.11), BMI (R^2^ = 0.12) and percentage weight gain (R^2^ = 0.11) during lockdown. Together, these food groups were good predictors of self-reported weight gain (Table 3). Conversely, in an univariate way, greater self-reported intake of raw and/or cooked vegetables and natural fruits (excluding juices) were associated with lower self-reported weight (Vegetables: B = −1.41, *p* < 0.001, Fruits: B = −1.19, *p* = 0.002), BMI (Vegetables: B = −0.51, *p* < 0.001, Fruits: B = −0.44, *p* = 0.001) and percentage weight gain during lockdown (Vegetables: B = −1.67, *p* = 0.002, Fruits: B = −1.61, *p* = 0.002) (Appendix A).

In the multilevel logistic regression models to determine the odds of experiencing a self-reported weight gain ≥ 5% during lockdown according to the frequencies of dietary consumption during lockdown, we observed that self-reported weight gain was higher with more frequent consumption of industrialized foods: 4–6 times a week (processed meats and sausages: OR = 1.99, 95%CI: 1.07–3.69, *p* = 0.029; sweet or filled cookies, cakes, etc: OR = 5.81, 95%CI: 2.10–16.09, *p* = 0.001; chocolates and chocolate-based products: OR = 3.42, 95%CI: 1.58–7.39, *p* = 0.002; salty snacks: OR = 5.25, 95%CI: 2.40–11.47, *p* < 0.001; drinks or juices whit added sugar: OR = 1.92, 95%CI: 2.40–11.47, *p* < 0.001) (Appendix A). Similarly, we analysed the different demographic and sedentary factors (Appendix A), lifestyle and eating habits during lockdown (Appendix A), and higher self-reported dietary intake during lockdown (Appendix A) to determine the main factors associated with ≥5% self-reported body weight gain during lockdown. In the multivariable model, we observed that differences in lifestyle, worsening eating habits, and more self-reported consumption of sweet or stuffed cookies and cakes were the main factors associated with self-reported weight gain during lockdown (Goodness of fit: Hosmer–Lemeshow *X^2^* = 9.992, *p* = 0.266; AUC: 0.75, 95%CI: 0.71–0.79, *p* < 0.001) (Table 4).

We determined the main dietary factors associated with self-reported weight gains of 5–9% or ≥ 10% during lockdown using a multinomial logistic regression model, observing that industrialised products, (sweet or filled cookies, cakes, etc; chocolates and chocolate-based products; salty snacks; drinks or juices with added sugar), were the main factors associated with self-reported weight gain. In contrast, higher self-reported consumption of processed meats and sausages, as well as rice, potatoes, noodles, or quinoa were only associated with higher odds of having a self-reported weight gain ≥ 10% during lockdown (Table 5).

Since differences in lifestyle and emotional feeding behaviour were the main factors associated with self-reported weight gain of ≥ 5% in the univariate logistic regression models (Appendix A), we created a random effects model to determine the differences in self-reported weight, BMI, and percentage weight gain during lockdown. We found that higher self-reported weight during lockdown was explained by differences in lifestyle regardless of emotional feeding behaviour; subjects with self-reported deterioration in their lifestyle had a greater self-reported weight and percentage weight gain than those who did not reported lifestyle differences or who had self-reported improvements during lockdown. Similar findings for BMI notwithstanding, if there was an interaction between lifestyle differences and emotional feeding behaviour, only 2% of differences in BMI were explained by this interaction, whereas 64% of the differences were due to a self-reported deterioration in lifestyle during lockdown (Figure 1).

## 5. Discussion

There is currently little evidence of the impact of lockdowns on weight variations, food consumption, and body weight. In this cross-sectional analytical study consisting of an online survey, we sought to evaluate differences in dietary behaviours and lifestyles on self-reported weight gain of students, academics, and administrative members of the University of Bío-Bío during the COVID-19 pandemic-derived lockdown in Chile.

Our study differed from others with a similar study design in that we were interested not only in differences in lifestyles, including eating habits, but also weight variations and the factors associated with self-reported body weight gain during lockdown. We found that the average self-reported weight gain during lockdown was 1.99 (SE: 0.17) kg and 0.7 (SE: 0.06) BMI units. These differences are greater than would be expected as a result of seasonal fluctuations, since even considering that the present study was carried out during winter in the southern hemisphere (21 July to 21 August), seasonal variations have only been accountable for increases of up to 0.7 kg and 0.5 units of BMI [22,23].

As other authors have reported, weight gain is often the result of the increase in the amount of food eaten, but particularly of those rich in simple carbohydrates and saturated fats [6,24,25,26]. For this reason, we developed three models for predicting differences in self-reported weight, BMI, and percentage weight gain. We observed that the differences were mainly associated with consumption of these foods together with processed foods, a situation that could potentially be circumvented with a higher consumption of fruits and vegetables [8,27]. This is important since weight losses of 5% may improve productivity and prevent the deterioration of health-related quality of life [28]. Furthermore, we found that the frequency of consumption of industrialized foods (cookies and sweet or filled cakes) was associated with the odds of self-reported weight gain during lockdown.

The main factors associated with a self-reported weight gain ≥ 5% during lockdown were lifestyle differences and emotional feeding behaviour. Self-reported weight gain during lockdown was related to differences in lifestyle and, unlike other studies, it was independent of emotional feeding behaviour [29,30,31]. Differences in self-reported weight, BMI, and percentage body weight were present to a greater extent in subjects with a self-reported worsened lifestyle, and, although emotional feeding behaviour was associated with a greater BMI, most of the association was related to differences in lifestyle during lockdown. Other studies have found that increased sedentary time, disturbed sleep, time spent on social media, and increased working time at home are associated with weight gain during lockdown [12,13,15].

When we evaluated dietary factors associated with self-reported weight gains greater than or equal to 10%, we observed that the determining associated factors were the consumption of industrialized foods, but unlike those observed for self-reported weight gains of 5–9%, the consumption of starchy foods such as rice, potatoes, noodles, or quinoa were aggregated associated factors. Weight gains of 10% are able to modify the brain, which may cause alterations in personality, leading to impulsiveness and lesser resistance to desires [32].

Other studies have evaluated eating behaviours and weight gain prior to the COVID-19 pandemic in similar Chilean populations. Pacheco and cols. studied emotional eating behaviour and cognitive restraint, both of which were associated with increased body weight and obesity in young adults [33]. Emotional eating has been reported to be highly prevalent, leading to excess eating in Chilean university students [34].

Lockdowns are local or regional interventions that involve people remaining confined in their households as long as possible, under new socially restrictive norms [35]. Lockdowns have been observed to cause loss of habits and routines, as well as psychosocial stress [36] that can trigger unhealthy habits like disordered or unhealthy eating, ceasing physical activity, altered sleep patterns, or worsening control of existing chronic diseases [8,9,11,37].

There are currently no sufficiently proven interventions for COVID-19 that may decrease mortality and complication rates at every stage of disease. Comorbidities like obesity, hypertension and diabetes mellitus worsen the prognosis of the disease [38,39,40,41]. Furthermore, disruptions in the gut microbiome have been hypothesized to be associated with disease progression in COVID-19 [42]. Thus, understanding how changes in lifestyle and eating habits impact distinct populations during lockdown is important to envision strategies that could prevent weight gain attributable to the aforementioned.

The main strength of this study is that it is one of the first studies evaluating the potential role of multiple exposures (eating habits before and during lockdown, emotional feeding behaviour, lifestyle before and during lockdown, and food safety) that may lead to self-reported weight gains between 5–9% and even greater than 10% during lockdown. Another strength is that our study was carried out approximately four months after the state of catastrophe was declared in Chile, which is when people suffered greater distress and fatigue due to prolonged confinement.

The main limitation of this study is that the survey was self-reported, which could have led to recall bias. Furthermore, weight was not measured directly before and during confinement by trained observers, being self-reported by respondents. However, these limitations could be nearly impossible to overcome when considering the challenges of conducting such a study during pandemic lockdown in a similar target population of people confined in their households. In addition, we did not assess emotional stress in our study, which could be an important confounder due to its potential role in modifying feeding behaviour and changes in lifestyle; future studies could seek to include emotional stress before and during confinement as one of the main variables. Another limitation is that we did not collect data of prior dietary and lifestyle regimens in participants who were healthy or had metabolic diseases before lockdown and whether these were maintained or not. Our study is representative of university academics, students, and administrative workers from a single region of Chile, thus, further characterisation of the impact of lockdowns in different populations is granted to guide decision-making. The amount of statistical analysis performed should be taken into consideration when interpreting these results since it could increase the probability that some of the findings could be due to chance. Furthermore, there was a low share of men in our study which reflects a potential bias in our sample.

Our study sheds light on some of the factors associated with self-reported weight gain during lockdown, which could be further studied in order to establish their role in weight gain during lockdowns in order to be considered by health authorities, as well as nutrition personnel to set dietary recommendations to follow during lockdown, including foods that should be avoided. Strategies based on these findings could be directed to limit the impact of differences in lifestyle and dietary behaviours during current and future lockdowns due to infectious disease outbreaks.

## 6. Conclusions

In this study, we have evaluated pre-lockdown and during-lockdown differences in lifestyle, eating habits, emotional feeding behaviour, and food safety, as well as their association with self-reported weight gain during lockdown. We characterised the factors associated with self-reported pre-lockdown vs during-lockdown total body weight gain between 5–9% and greater than or equal to 10%. Differences in dietary behaviours during lockdown, mainly higher reported consumption of industrialized foods, as well as quality of life differences were the main factors associated with self-reported weight gain during the COVID-19 lockdown in Chile.

## Figures and Tables

**Figure 1 nutrients-13-03213-f001:**
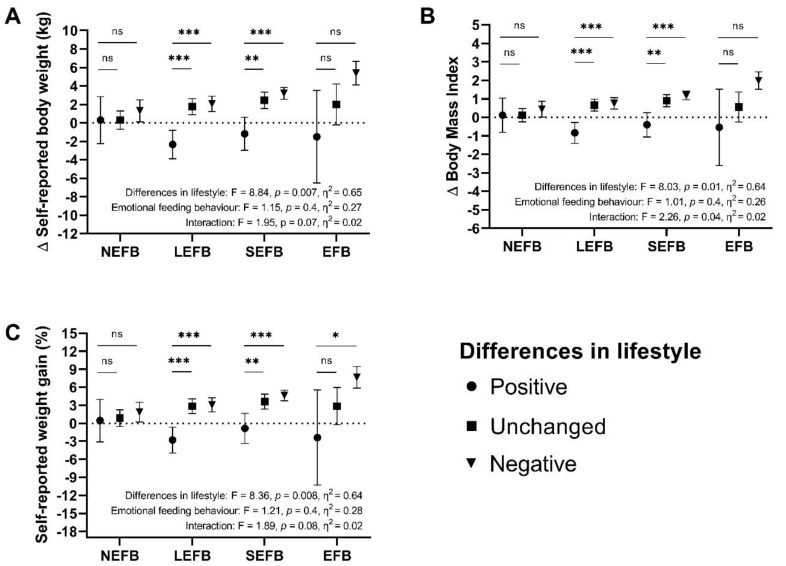
Results of the random effects models for the differences in self-reported (**A**). body weight, (**B**). BMI, and (**C**). percentage weight gain for differences in lifestyle adjusted for emotional feeding behaviour. Mean and 95%CI are shown. NEFB: No emotional eating behaviour, LEFB: Little emotional eating behaviour, SEFB: Some emotional eating behaviour, EFB: Emotional eating behaviour, Δ: Difference, BMI: Body Mass Index, ns: No significance. *: *p* < 0.05, ** *p* < 0.01, *** *p* < 0.001.

**Table 1 nutrients-13-03213-t001:** Descriptive characteristics of the population, disaggregated by sex.

	Total *n* = 639		Women *n* = 427		Men *n* = 212		*p* Value
	Mean	Standard Deviation	Mean	Standard Deviation	Mean	Standard Deviation	
Anthropometric Data							
Age, years	28.9	13.2	28.5	12.4	29.9	14.6	0.226
BMI before lockdown	25.1	4.8	25.1	5.2	25.1	4.1	0.897
BMI during lockdown	25.8	5.0	25.9	5.3	25.8	4.4	0.847
Sociodemographic and feeding behaviour data							
	Frequency (percentage)		Frequency (percentage)		Frequency (percentage)		
Place of residence							
*Urban*	546 (85.4)		361 (84.5)		185 (87.3)		0.405
*Rural*	93 (14.6)		66 (15.5)		27 (12.7)		
Occupation							
*Student*	461 (72.1)		310 (72.6)		151 (71.2)		0.887
*Academic*	84 (13.1)		51 (11.9)		33 (15.6)		
*Administrative*	94 (14.7)		66 (15.5)		28 (13.2)		
Educational level							
*Basic (incomplete)*	4 (0.6)		3 (0.7)		1 (0.5)		0.105
*Basic (complete)*	30 (4.7)		19 (4.4)		11 (5.2)		
*Technical (incomplete)*	6 (0.9)		5 (1.2)		1 (0.5)		
*Technical (complete)*	26 (4.1)		23 (5.4)		3 (1.4)		
*University (incomplete)*	405 (63.4)		272 (63.7)		133 (62.7)		
*University (complete)*	71 (11.1)		49 (11.5)		22 (10.4)		
*Postgraduate*	97 (15.2)		56 (13.1)		41 (19.3)		
Telecommuting from home	622 (97.3)		416 (97.4)		206 (97.2)		0.851
Time for sedentary activities							
*One to two hours a day*	32 (5)		21 (4.9)		11 (5.2)		0.837
*Three to four hours a day*	70 (11)		39 (9.1)		31 (14.6)		
*Five to six hours a day*	109 (17.1)		78 (18.3)		31 (14.6)		
*Seven to eight hours*	161 (25.2)		112 (26.2)		49 (23.1)		
*Nine o ten hours a day*	108 (16.9)		77 (18)		31 (14.6)		
*Ten or more hours a day*	159 (24.9)		100 (23.4)		59 (27.8)		
Household members							
*Lives alone*	28 (4.4)		21 (4.9)		7 (3.3)		0.139
*Lives with family (parents and/or siblings or partner and children)*	529 (82.8)		357 (83.6)		172 (81.1)		
*Lives with relatives who are not parents and siblings*	28 (4.4)		17 (4)		11 (5.2)		
*Lives with friends*	7 (1.1)		5 (1.2)		2 (0.9)		
*Lives with other people who are not family members*	9 (1.4)		4 (0.9)		5 (2.4)		
*Lives with partner*	35 (5.5)		21 (4.9)		14 (6.6)		
*Other*	3 (0.5)		2 (2)		1 (0.5)		
Lunch	365 (57.1)		232 (54.3)		133 (62.7)		0.043
Away from home	297 (81.4)		199 (85.8)		98 (73.7)		0.928
Packed lunch for work	206 (69.4)		159 (79.9)		47 (47.9)		< 0.001
Differences in habits							
*Yes, it has gotten worse*	268 (41.9)		191 (44.7)		77 (36.3)		0.004
*Yes, it has improved*	190 (29.7)		133 (31.1)		57 (26.9)		
*No, it has stayed the same*	181 (28.3)		103 (24.1)		78 (36.8)		
Mealtimes before the pandemic							
*Breakfast*	521 (81.5)		348 (81.5)		173 (81.6)		0.974
*Mid-morning snack*	263 (41.2)		200 (46.8)		63 (29.7)		<0.001
*Lunch*	613 (95.9)		409 (95.8)		204 (96.2)		0.790
*Mid-afternoon snack*	291 (45.5)		208 (48.7)		83 (39.2)		0.022
*Snack before dinner*	517 (80.9)		351 (82.2)		166 (78.3)		0.238
*Dinner*	141 (22.1)		90 (21.1)		51 (24.1)		0.392
*Late-night snack*	119 (18.6)		70 (16.4)		49 (23.1)		0.040
*Snacking between meals*	219 (34.3)		151 (35.4)		68 (32.1)		0.410
Mealtimes during the pandemic							
*Breakfast*	504 (78.9)		335 (78.5)		169 (79.7)		0.713
*Mid-morning snack*	164 (25.7)		111 (26)		53 (25)		0.786
*Lunch*	623 (97.5)		416 (97.4)		207 (97.6)		0.868
*Mid-afternoon snack*	294 (46)		214 (50.1)		80 (37.7)		0.003
*Snack before dinner*	529 (82.8)		363 (85)		166 (78.3)		0.034
*Dinner*	163 (25.5)		98 (23)		65 (30.7)		0.035
*Late-night snack*	214 (33.5)		144 (33.7)		70 (33)		0.859
*Snacking between meals*	319 (49.9)		219 (51.3)		100 (47.2)		0.327
Snack between meals							
*Does not snack between meals*	95 (14.9)		58 (13.6)		37 (17.5)		0.614
*Same as before*	120 (18.8)		83 (19.4)		37 (17.5)		
*More than before*	282 (44.1)		190 (44.5)		92 (43.4)		
*Less than before*	142 (22.2)		96 (22.5)		46 (21.7)		
Emotional feeding behaviour							
*No emotional feeding behaviour*	121 (18.9)		59 (13.8)		62 (29.2)		<0.001
*Little emotional feeding behaviour*	204 (31.9)		131 (30.7)		73 (34.4)		
*Some emotional feeding behaviour*	260 (40.7)		191 (44.7)		69 (32.5)		
*Emotional feeding behaviour*	54 (8.5)		46 (10.8)		8 (3.8)		
Lifestyle before the pandemic							
*Fantastic lifestyle*	76 (11.9)		47 (11)		29 (13.7)		0.251
*Right path*	306 (47.9)		202 (47.3)		104 (49.1)		
*Adequate*	164 (25.7)		114 (26.7)		50 (23.6)		
*Could be better*	89 (13.9)		61 (14.3)		28 (13.2)		
*Danger zone*	4 (0.6)		3 (0.7)		1 (0.5)		
Lifestyle during the pandemic							
*Fantastic lifestyle*	26 (4.1)		13 (3)		13 (6.1)		0.196
*Right path*	177 (27.7)		119 (27.9)		58 (27.4)		
*Adequate*	167 (26.1)		102 (23.9)		65 (30.7)		
*Could be better*	242 (37.9)		181 (42.4)		61 (28.8)		
*Danger zone*	27 (4.2)		12 (2.8)		15 (7.1)		
Food Safety							
*Safe*	222 (34.7)		148 (34.7)		74 (34.9)		0.315
*Mildly unsafe*	265 (41.5)		185 (43.3)		80 (37.7)		
*Moderately unsafe*	86 (13.5)		54 (12.6)		32 (15.1)		
*Severely unsafe*	66 (10.3)		40 (9.4)		26 (12.3)		

Body mass index (BMI). Comparisons were made with Student’s *t* test for independent sample, X^2^, and X^2^ for trend. Subcategories for every main variable are shown in italics.

**Table 2 nutrients-13-03213-t002:** Comparison of the difference in weight, BMI, and percentage weight before and during lockdown between subjects with the same self-reported intake or higher self-reported intake by food groups.

	Weight Difference				BMI Difference				Percentage Weight Gain			
	Same Self-Reported Intake		Higher Self-Reported Intake		Same Self-Reported Intake		Higher Self-Reported Intake		Same Self-Reported Intake		Higher Self-Reported Intake	
	Mean	Standard Error	Mean	Standard Error	Mean	Standard Error	Mean	Standard Error	Mean	Standard Error	Mean	Standard Error
White or whole wheat bread	1.68	0.19	2.92 **	0.32	0.61	0.07	1.08 **	0.11	2.61	0.27	4.27 **	0.45
Rice, potatoes, noodles, or quinoa	1.59	0.21	2.85 ***	0.29	0.58	0.07	1.06 ***	0.11	2.45	0.28	4.25 ***	0.41
Raw and/or cooked vegetables	2.33	2.33	0.93 **	0.38	0.85	0.06	0.35 **	0.14	3.43	0.26	1.76 **	0.52
Natural fruit (excludes juices)	2.32	2.32	1.13 **	0.37	0.85	0.06	0.41 **	0.14	3.47	0.26	1.86 **	0.50
Dried vegetables	2.14	0.19	1.68	0.33	0.78	0.07	0.62	0.12	3.21	0.27	2.61	0.45
Milk, yogurt, or kefir	2.18	0.18	1.46	0.36	0.80	0.06	0.53	0.13	3.27	0.26	2.34	0.49
Cheeses (aged, fresh, farm, etc.)	1.79	0.19	2.66 *	0.35	0.65	0.07	0.98 *	0.13	2.75	0.26	4.04 *	0.49
Meat (pork, chicken, beef, lamb, etc.)	1.85	0.19	2.66	0.36	0.68	0.06	0.97	0.13	2.78	0.26	4.15 *	0.50
Processed meats and sausages	1.65	0.18	3.43 ***	0.36	0.60	0.06	1.26 ***	0.12	2.55	0.26	4.98 ***	0.47
Fresh and canned seafood	2.07	0.18	1.76	0.36	0.76	0.06	0.64	0.13	3.09	0.26	2.82	0.50
Eggs	1.97	0.18	2.05	0.37	0.72	0.06	0.75	0.13	2.97	0.26	3.19	0.50
Nuts (excludes raisins)	2.08	0.18	1.65	0.41	0.75	0.06	0.62	0.14	3.11	0.26	2.71	0.54
Butter, margarine, vegetable oil or fats of animal origin	1.77	0.21	2.64 *	0.28	0.65	0.07	0.96 *	0.10	2.73	0.27	3.88 *	0.42
Sweet or filled cookies, cakes, etc.	1.33	0.19	3.59 ***	0.31	0.48	0.07	1.33 ***	0.11	2.07	0.26	5.34 ***	0.43
Chocolates & chocolate-based products	1.46	0.19	3.43 ***	0.30	0.53	0.07	1.26 ***	0.11	2.23	0.27	5.16 ***	0.42
Salty snacks	1.36	0.19	3.59 ***	0.29	0.49	0.07	1.32 ***	0.11	2.15	0.27	5.23 ***	0.41
Drinks or juices with added sugar	1.57	0.19	3.49 ***	0.31	0.57	0.07	1.30 ***	0.11	2.39	0.26	5.26 ***	0.45
Alcoholic drinks	1.87	0.18	2.61	0.40	0.68	0.06	0.69	0.15	2.84	0.26	3.98	0.54
Sugar (white or brown)	1.68	0.19	3.37 ***	0.31	0.61	0.07	1.26 ***	0.11	2.63	0.26	4.81 ***	0.44

Comparisons were made with Student’s *t* test for independent samples. The mean values of the subjects who self-reported a higher intake during lockdown are significantly different from those with the same self-reported intake: * *p* < 0.05, ** *p* < 0.01, *** *p* < 0.001.

**Table 3 nutrients-13-03213-t003:** Multivariable linear regression models for the prediction of differences of self-reported weight, BMI, and percentage weight percentage from more self-reported food intake during lockdown.

	Regression Coefficients	Standard Error	*p* Value
Model for weight			
Constant	1.38	0.45	0.002
More consumption of sweet or filled cookies, cakes, etc.	1.32	0.39	0.001
More consumption of salty snacks	1.22	0.40	0.002
More consumption of processed meats and sausages	0.85	0.43	0.044
More sugar consumption (white or brown)	0.87	0.43	0.030
More consumption of rice, potatoes, noodles or quinoa	0.78	0.35	0.027
Model for BMI			
Constant	0.53	0.16	0.001
More consumption of sweet or filled cookies, cakes, etc.	0.49	0.15	0.001
More consumption of salty snacks	0.45	0.15	0.002
More sugar consumption (white or brown)	0.35	0.16	0.027
More consumption of rice, potatoes, noodles or quinoa	0.30	0.13	0.021
More consumption of processed meats and sausages	0.32	0.16	0.038
Model for percentage weight gain			
Constant	2.55	0.62	<0.001
More consumption of sweet or filled cookies, cakes, etc.	2.12	0.55	<0.001
More consumption of salty snacks	1.48	0.58	0.010
More consumption of sugary drinks or juices	1.53	0.59	0.009
More consumption of rice, potatoes, noodles or quinoa	1.19	0.49	0.015

Models adjusted for sex (man), age (years), and occupation (academic and administrative).

**Table 4 nutrients-13-03213-t004:** Multivariable logistic regression model for predicting ≥5% self-reported body weight gain during lockdown.

	B	Standard Error	OR	95%CI	*p* Value
Constant	−2.62	0.80	-	-	0.001
Lifestyle during the pandemic					
Fantastic lifestyle	Reference				
Right path	1.09	0.77	2.98	(0.66–13.42)	0.154
Adequate	1.29	0.77	3.64	(0.81–16.41)	0.093
Could be better	1.42	0.77	4.12	(0.92–18.57)	0.065
Danger zone	2.65	0.91	14.21	(2.35–85.82)	0.004
Worsening eating habits	1.23	0.20	3.43	(2.31–5.09)	<0.001
More consumption of sweet or stuffed cookies, cakes, etc.	0.75	0.20	2.11	(1.42–3.13)	<0.001

B: Regression coefficient, OR: Odds ratio, 95%CI: 95% confidence interval. Models adjusted for sex (man), age (years), and occupation (academic and administrative).

**Table 5 nutrients-13-03213-t005:** Results of the multinomial logistic regression analysis to determine the odds of 5–9% or ≥10% self-reported weight gain due to higher self-reported dietary intake during lockdown.

	5–9% Weight Gain			≥10% Weight Gain		
	OR	95%CI	*p* Value	OR	95%CI	*p* Value
White or whole wheat bread	1.19	0.79–1.83	0.397	1.73	0.99–3.02	0.053
Rice, potatoes, noodles, or quinoa	1.41	0.96–2.08	0.080	2.26	1.33–3.82	0.003
Raw and/or cooked vegetables	0.88	0.57–1.34	0.544	0.61	0.32–1.17	0.137
Natural fruit (excludes juices)	0.89	0.59–1.34	0.570	0.60	0.32–1.12	0.108
Dried vegetables	0.94	0.63–1.39	0.757	0.89	0.51–1.56	0.688
Milk, yogurt, or kefir	1.13	0.75–1.70	0.564	0.75	0.40–1.39	0.351
Cheeses (aged, fresh, etc.)	1.41	0.93–2.15	0.107	1.21	0.66–2.21	0.541
Meat (pork, chicken, beef, lamb, etc.)	1.21	0.75–1.95	0.428	1.66	0.90–3.08	0.105
Processed meats and sausages	1.53	0.97–2.41	0.068	2.61	1.46–4.66	0.001
Fresh and canned seafood	0.87	0.57–1.33	0.520	1.19	0.67–2.10	0.554
Eggs	1.12	0.74–1.71	0.595	1.21	0.68–2.17	0.518
Nuts (excludes raisins)	0.94	0.60–1.48	0.792	1.00	0.54–1.88	0.990
Butter, margarine, vegetable oil or fats of animal origin	1.28	0.85–1.93	0.234	1.21	0.68–2.16	0.521
Sweet or filled cookies, cakes, etc.	3.23	2.16–4.76	<0.001	3.61	2.10–6.21	<0.001
Chocolates and chocolate-based products	3.25	2.18–4.85	<0.001	2.32	1.32–4.07	0.003
Salty snacks	2.89	1.95–4.29	<0.001	2.78	1.61–4.80	<0.001
Drinks or juices with added sugar	2.01	1.31–3.07	0.001	2.39	1.35–4.23	0.003
Alcoholic drinks	1.69	1.04–2.73	0.034	1.43	0.69–2.97	0.330
Sugar (white or brown)	1.40	0.89–2.22	0.150	1.56	0.84–2.92	0.162

OR: Odds ratio, 95%CI: 95% confidence intervals. Models adjusted for sex (man), age (years), and occupation (academic and administrative).

## Data Availability

The data presented in this study are available on request from the corresponding author.

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
