# Peer review of "Association of Differences in Dietary Behaviours and Lifestyle with Self-Reported Weight Gain during the COVID-19 Lockdown in a University Community from Chile: A Cross-Sectional Study"

_nutrients, 2021, doi:10.3390/nu13093213_

Round 1

Reviewer 1 Report

The manuscript entitled “Association of differences in dietary behaviours and lifestyle with self-reported weight gain during the COVID-19 lockdown in a university community from Chile: A cross-sectional study” presents interesting issue. I appreciate the great efforts that the authors have made in response to my previous questions and concerns (previous review – before resubmission). However, there are some issues that should be corrected:

  • Line 99 – „from July 21 to August 19” – please add the year
  • Figure 1 should be improved (it is difficult to read)

Author Response

Dear editors and reviewers. We would like to thank you once more for your effort and time spent in reviewing our manuscript. These are our individual responses to your comments:

Reviewer 1

[…] there are some issues that should be corrected:

  1. Line 99 – „from July 21 to August 19” – please add the year

R: Thank you, we have added the year as suggested.

Figure 1 should be improved (it is difficult to read)

R: We have redesigned Figure 1 by substituting the long labels for every column with abbreviations. We have also ordered in a different order all three panels to form a squared figure which allows to display dis figure in a bigger size with better visualization of all elements.

Reviewer 2 Report

In this manuscript, the authors explored the association of differences in dietary behaviors and lifestyle with self-reported weight gain during the COVID-19 lockdown in Chile. This research is significant to understand the impact of lockdown on eating habits and lifestyle, as well as weight gain. The study itself is well conducted but some concerns arise as following:

  1. Introduction: the authors firstly explained the impact of lockdown on health behaviors and lifestyles, which might be caused by a high prevalence of psychological distress. Then they listed some studies on associations of changes in weight and lifestyle during the COVID-19 lockdown. But the authors did not clearly explain the significance of exploring weight gain and its risk factors during lockdown. I agree that understanding of how lockdowns are associated with different lifestyles and dietary behaviors is important to limit their adverse effects on health, but why the authors focus on weight gain rather than obesity or other chronic diseases which are more harmful to health?
  2. In page 2 lines 79-80, relevant references should be cited for the “five surveys”.
  3. Did the authors analyze the reliability and validity of the condensed questionnaires to ensure the accuracy of the survey?
  4. How to control the quality of online survey? and potential bias?
  5. In page 2 lines 86-88, the exclusion criteria did not consider the people with chronic disease who might manage their disease by controlling their dietary behavior and lifestyle autonomously, which could confuse the analysis results.
  6. Please explain the “emotional feeding behavior” in short words for better understanding its significance in this research.
  7. In page 3 lines 102-104, it is not clear that whether the weight used in the final analysis before confinement and during lockdown are weight self-reported only once or the average values calculated from several self-reports.
  8. In page 3 line 109, references should be cited for the “CFCA”.
  9. In page 3 lines 108-112, the authors used CFCA to measure different foods consumption, but only foods consumption frequencies were used in the final analysis, why the authors did not take the average daily food consumption into account?
  10. How to define the lower during-lockdown food intake and the higher during-lockdown food intake?
  11. Is there some differences between the pre-lockdown CFCA and the during-lockdown CFCA?
  12. In page 3 lines 143-144, authors mentioned the test of normality, but they did not further explain the impact of the test results on the selection of statistical description and inference methods. If the distribution is skewed, the t-test is not suitable. Or whether all the normality tests results are normal distributions? The authors need to make it clear.
  13. In page 4 lines 153-154, the sentence of “since consumption of specific food groups is not necessarily interdependent to the consumption of others” is confusing. As we all know, an individual’s daily energy intake is typically consumed with certain constraints, so that different groups of foods consumption can not be independent of each other. The relationship between different foods consumption should be a trade-off, because the sum of foods intake is limited.
  14. Specify how covariates were modeled in the multivariable models (e.g. continuous, categorical, what categories).
  15. In page 4 line 170 and page 10 line 236, “a multinomial regression model” should be “a multinomial logistic regression model”.
  16. The section of statistical analysis is lengthy, authors should simplify this part.
  17. Results: when describing the population characteristics, why the population is divided into different groups by the indicator of sex rather than the indicator of self-reported weight gain or not during lockdown.
  18. The authors reported an increase in BMI during lockdown, but what about the percentages of overweight and obesity in the population?
  19. Discussion: I wonder whether there are other studies have evaluated the association between different foods consumption and weight gain without the spread of COVID-19 or before lockdown. The authors should compare their findings with those from other studies before the spread of COVID-19 or lockdown.

Author Response

Dear editors and reviewers. We would like to thank you once more for your effort and time spent in reviewing our manuscript. These are our individual responses to your comments:

Reviewer 2

  1. Introduction: the authors firstly explained the impact of lockdown on health behaviors and lifestyles, which might be caused by a high prevalence of psychological distress. Then they listed some studies on associations of changes in weight and lifestyle during the COVID-19 lockdown. But the authors did not clearly explain the significance of exploring weight gain and its risk factors during lockdown. I agree that understanding of how lockdowns are associated with different lifestyles and dietary behaviors is important to limit their adverse effects on health, but why the authors focus on weight gain rather than obesity or other chronic diseases which are more harmful to health? 

R: It is true that obesity and other chronic diseases are more harmful to health than weight gain. Also, with our current understanding of the pandemic, these outcomes have gained relevance to be studied in the context of the pandemic since the pandemic has extended for almost two years now and lockdowns will most probably have long-term effects on the health of our populations. However, when this study was conceived at the beginning of 2020, the generalized thinking was that the COVID-19 associated lockdown would be short-lived and, thus, we considered that evaluating a short-term outcome like weight gain was more appropriate than seeking to evaluate long-term outcomes. Furthermore, short-term changes in weight associated with short-term overfeeding are associated with negative effects on metabolism, which justifies studying changes in weight and dietary habits in the short-term. We have added this in the introduction of our paper to expand the rationale for studying factors associated with weight gain during lockdown.

  1. In page 2 lines 79-80, relevant references should be cited for the “five surveys”.

R: Thank you. We have now included all relevant references for the five surveys and we have also numbered them in their first appearance in the manuscript for greater clarity.

  1. Did the authors analyze the reliability and validity of the condensed questionnaires to ensure the accuracy of the survey?

R: All these questionnaires had been previously validated and applied in the Chilean population, including nationwide population-based studies, as well as other populations. What we did was assemble these questionnaires into a single google forms document consisting of 85 items. Afterwards, we asked for this assembled questionnaire to be evaluated by as many experts as possible from the Department of Nutrition and Public Health from the University of Bío-Bío. Thus, this assembled questionnaire was validated by experts in the field. Regarding individual validations of the questionnaires and their applications in Spanish and in the Chilean population:

  1. The Food Consumption Frequency Questionnaire was taken from the questionnaire applied in the National Food Consumption Study from the Ministry of Health of Chile which was conducted in the Chilean population in Spanish in the year 2010. References:
    • Ministerio de salud, Gobierno de chile. Encuesta nacional de consumo alimentario. Chile: Facultad de medicina-Universidad de Chile Economía y Negocios-Universidad de Chile; 2014.p166-191. Available in: http://www.minsal.cl/sites/default/files/ANEXOS_ENCA.pdf
    • Herrera Figueroa Y. Norma general Técnica N° 148/2013 sobre Guías Alimentarias (GABA) para la Población Chilena, aprobada por Resolución N° 260/2013 del MINSAL. Santiago, Chile: Ministerio de salud; 2013.
  2. The Emotional Eating Survey was originally developed in Spanish and applied in the Spanish population. This survey was posteriorly validated in the Chilean population. References:
    • Garaulet M, Canteras M, Morales E, López-Guimera G, Sánchez-Carracedo D, Corbalán-Tutau MD. Validación de un cuestionario sobre alimentación emocional para su uso en casos de obesidad: el Cuestionario Eater emocional (EEQ). Nutr Hosp . 2012; 27 (2): 645-651. doi: 10.1590 / S0212-16112012000200043
    • Gonzalez Mariela Validación del Cuestionario de Comedor Emocional (CCE) en Chile Revista GEN (Gastroenterología Nacional) 2018;72(1):21-24.
  3. The Fantastic Questionnaire was created by the Ministry of Health of Chile to measure lifestyles in the general population in an easy-to-answer manner. Thus, it was originally developed in Spanish and validated by both the Ministry of Health of Chile and other authors from Latin America. References:
    • ¿Cómo es tu estilo de vida? [Internet]. Buenas Prácticas APS. 2013 [cited 8 August 2020]. Available from: http://buenaspracticasaps.cl/wp-content/uploads/2013/10/Como_es_tu_estilo_de_vida.pdf
    • Betancurth Loaiza, Diana Paola, & Vélez Álvarez, Consuelo, & Jurado Vargas, Liliana (2015). Validación de contenido y adaptación del cuestionario Fantastico por técnica Delphi. Salud Uninorte, 31(2),214-227.
    • Ramírez-Vélez. R. & Agredo. R.A. (2012). Fiabilidad y validez del instrumento “Fantástico” para medir el estilo de vida en adultos colombianos. Revista Salud Pública, 14 (2), 226-237, 2012.
    • Barriga Silva, T. A. (2020). Instrumento "Fantástico" para medir el estilo de vida saludable de adolescentes de la comuna de Bulnes. Revista Reflexión E Investigación Educacional, 3(1), 61-74.
  4. The Household Food Insecurity Access Component Scale was originally developed in Spanish with cooperation form the Global Health Bureau of the United States Agency for International Development, to be applied in developing countries. This questionnaire has been extensively validated. References:
    • Coates, Jennifer, Anne Swindale y Paula Bilinsky. Escala del Componente de Acceso de la Inseguridad Alimentaria en el Hogar (HFIAS) para la Medición del Acceso a los Alimentos en el Hogar: Guía de Indicadores (v. 2). Washington, D.C.: Proyecto de Asistencia Técnica sobre Alimentos y Nutrición, Academia para el Desarrollo Educativo, agosto de 2007
    • Salvador G, Ngo J, Pérez C, Aranceta J. Escalas de evaluación de la inseguridad alimentaria en el hogar. Rev Esp Nutr Comunitaria. 2015;21: 270-276.
    • Ballard, T., Kepple, A., & Cafiero, C. (2013). The Food Insecurity Experience Scale. Development of a Global Standard for Monitoring Hunger Worldwide. (FAO, Ed.) Technical Paper. Rome, FAO.
    • Cafiero C, Melgar-Quiñonez HR, Ballard TJ, Kepple AW. Validity and reliability of food security measures. Ann N Y Acad Sci. 2014 Dec; 1331:230-248. doi: 10.1111/nyas.12594.
    • Castell, G., De la Cruz, J., Pérez, C., & Aranceta, J. (2015). Escalas de evaluación de la inseguridad alimentaria en el hogar. Revista Española de Nutrición, 21(1), 270-276.
    • Féliz-Varduzco, G., Aboites-Manrique, G., & Castro-Lugo, D. (2018). La seguridad alimentaria y su relación con la suficiencia e incertidumbre del ingreso: un análisis de las percepciones del hogar. Acta Universitaria, 28(4), 74-86

  1. How to control the quality of online survey? and potential bias?

R: All online surveys are subject to selection bias since there will always be a fraction of the potential participants who will decide not to participate in the survey. However, there are strategies to limit selection bias as much as possible. The first strategy is to clearly define the population and when defining the sample, give an equal opportunity to every potential participant to participate. In our case, we had a clearly defined objective population and we gave an equal opportunity to participate to every user of the university mailing system which limits selection bias. Also, participants were given more than one opportunity to participate throughout several weeks in order to reduce selection bias.

  1. In page 2 lines 86-88, the exclusion criteria did not consider the people with chronic disease who might manage their disease by controlling their dietary behavior and lifestyle autonomously, which could confuse the analysis results.

R: It is true that participants could have been under autonomous dietary and lifestyle control which would have been informative. We have now discussed this as a limitation of our manuscript. However, we do not consider that this could be a confounding variable since this questionnaire specifically assessed changes occurred during lockdown with respect to baseline regimens of dietary behaviours and lifestyle. Participants who reported having the same dietary regimen during lockdown were classified as unchanged dietary habits, whereas participants were asked if they increased or reduced consumption of distinct foods, as well as dietary habits. Similarly, changes in lifestyle were reported as compared to pre-lockdown lifestyle. Responses to these items, however, may have an implicit recall bias which is an important limitation which we had already recognized in the discussion of our manuscript.

  1. Please explain the “emotional feeding behavior” in short words for better understanding its significance in this research. 

R: Thank you. We have now included a brief definition for this according to the original reference.

  1. In page 3 lines 102-104, it is not clear that whether the weight used in the final analysis before confinement and during lockdown are weight self-reported only once or the average values calculated from several self-reports. 

R: These are self-reported weights as reported by participants. They do not correspond to averages from distinct self-reports. We have added a clarification in the corresponding section to clarify that these were self-reported by participants as independent measurements.

  1. In page 3 line 109, references should be cited for the “CFCA”.

R: Thank you. We have now added this reference.

  1. In page 3 lines 108-112, the authors used CFCA to measure different foods consumption, but only foods consumption frequencies were used in the final analysis, why the authors did not take the average daily food consumption into account? 

R: CFCA was designed to obtain frequencies of consumption of diverse foods which are regularly consumed by the Chilean population. It does not take into account the exact amounts for individual foods since it is difficult for both interviewers and responders of the questionnaire to provide precise daily or weekly amounts of consumption of these foods. This is why these kinds of questionnaires have been preferred with respect to 24-hour dietary recall questionnaires which are considered to be semi-exact since they investigate quantities but have a high probability of recall bias. Therefore, we decided to use CFCA due to its easy deliver through an online survey in the absence of expert interviewers who could have probably been able to obtain more precise reporting of quantities of foods.  

  1. How to define the lower during-lockdown food intake and the higher during-lockdown food intake?

R: Differences in dietary intake were defined according to self-reported frequency of consumption of foods during-lockdown, compared with that before lock-down, based on an ordinal scale including three possible options: lower, same, and higher. We have added this clarification in the manuscript.

  1. Is there some differences between the pre-lockdown CFCA and the during-lockdown CFCA?

R: In the prior version of the manuscript, we had included in the supplementary materials the frequencies of consumption of foods according to CFCA before and during-lockdown. In this update of our manuscript, we have included in Table 1 of supplementary materials, the corresponding frequencies, and we have added a statistical comparison of these frequencies; the p-value was obtained with the McNemar test to evaluate if there are differences between pre-lockdown and during-lockdown frequencies of consumption of foods.

  1. In page 3 lines 143-144, authors mentioned the test of normality, but they did not further explain the impact of the test results on the selection of statistical description and inference methods. If the distribution is skewed, the t-test is not suitable. Or whether all the normality tests results are normal distributions? The authors need to make it clear.

R: The distribution for quantitative variables was evaluated through kurtosis and asymmetry, the existence of a normal distribution was verified for all variables. Parametric tests were used since there were different criteria justifying their use, not only the presence of a normal distribution, but also the independence of samples and equality of variances. For the random effects model, normality of residues was evaluated to determine that the model was correct. We have further clarified in the manuscript that this is why parametric tests were used.

  1. In page 4 lines 153-154, the sentence of “since consumption of specific food groups is not necessarily interdependent to the consumption of others” is confusing. As we all know, an individual’s daily energy intake is typically consumed with certain constraints, so that different groups of foods consumption can not be independent of each other. The relationship between different foods consumption should be a trade-off, because the sum of foods intake is limited. 

R: Thank you, we agree with this comment and believe that it is misleading, while it does not provide essential information for overall understanding of the manuscript. We have therefore removed this sentence.

  1. Specify how covariates were modeled in the multivariable models (e.g. continuous, categorical, what categories).

R: Thank you, we have now clarified in the statistical analysis section that age was used as a continuous variable. We have also indicated all the variables used for adjustment in the footnotes of the tables.

  1. In page 4 line 170 and page 10 line 236, “a multinomial regression model” should be “a multinomial logistic regression model”.

R: Thank you, we have changed this accordingly in the text.

  1. The section of statistical analysis is lengthy, authors should simplify this part.

R= We have reduced this section to make it as concise as posible.

  1. Results: when describing the population characteristics, why the population is divided into different groups by the indicator of sex rather than the indicator of self-reported weight gain or not during lockdown.

R: Thank you for this suggestion. We have now added a table in the Supplementary Data (Supplementary Table 1) in which we compare the characteristics of participants according to self-reported percentage weight gain. We chose to group participants in this form since percentage weight gain is an objective way to group participants into well-defined and comparable categories.

The reason why we decided to group patients by sex and why we have decided to leave table 1 in our manuscript with data disaggregated by sex was that our sample was composed of a great proportion of women. For this reason, we considered important to show readers that data tended to be homogeneous between different sex, and thus, we did not have to perform sub analyses according to sex. Nonetheless, we consider that reporting data according to groups of weight gain is also important and informative, which is why we chose to include this new table.

  1. The authors reported an increase in BMI during lockdown, but what about the percentages of overweight and obesity in the population?

R= Thank you for this recommendation. We have now added in the results section of the manuscript the percentages of overweight and obesity before and during-lockdown in our study to reflect how these changed.

  1. Discussion: I wonder whether there are other studies have evaluated the association between different foods consumption and weight gain without the spread of COVID-19 or before lockdown. The authors should compare their findings with those from other studies before the spread of COVID-19 or lockdown.

R: Thank you for this suggestion. We have added a comparison with other studies which have evaluated factors associated to weight gain in the Chilean population prior to the pandemic.  

This manuscript is a resubmission of an earlier submission. The following is a list of the peer review reports and author responses from that submission.

Round 1

Reviewer 1 Report

The manuscript entitled “Changes in dietary behaviours and lifestyle as risk factors for 2 weight gain during the COVID-19 lockdown in Chile: A cross-3 sectional study” presents interesting issue, however some corrections are needed

  • Introduction - In this section Authors presented the information associated with the COVID-19 pandemic. However, some detailed information about global situation is missing. This section should be briefly presented – what do we know and what is the background for this study. Some detailed information about other studies are necessary. The good background should present the history of problem, the current knowledge and scientific "gap", and then authors should present how their study could fill this gap to justify the study. The international context must be presented. Authors presented issues that may have influenced weight gain, but at the same time, there are numerous studies published that indicated the opposite situation associated with following a better diet during lock down, that should be also indicated, e.g. https://www.ncbi.nlm.nih.gov/pmc/articles/PMC7278251/; https://www.mdpi.com/2072-6643/12/9/2640.
  • Line 77 - A sentence should never start with a number
  • Materials and methods – how the recruitment was carried out?  How the subjects were selected? Why participants were recruited only from the university community?
  • Food Consumption Frequency Questionnaire; Emotional Eating Survey; Fantastic questionnaire; Household Food Insecurity Access Component Scale - Was the questionnaire previously validated? What was the accuracy and consistency of this questionnaire? What is the original language of the questionnaire? Was the questionnaire translated? Who did so? Any validation of the translated questionnaire?
  • Lines 109-110 - Was the normality of distribution tested? The information about it should be added and authors should be consequent. If data have normal distribution, they should be treated as such, if not, nonparametric tests should be applied. Please specify it.
  • Table 1 – please remove infoamion about weight (please remain only the data of BMI). It is misleading – it is natural that the weight of men and women differ!
  • Tables are a little bit difficult to follow – please improve it.  
  • Please unify the number of decimal places in p-Values
  • Authors should in their discussion include 3 areas: (1) compare gathered data with the results by other authors, (2) formulate implications of the results of their study and studies by other authors, (3) formulate the future areas which should be studied.
  • Authors should present here and discuss the limitations of their study.
  • In limitation section (at the end of the discussion section) authors should add the information about low share of men in the sample (as a bias)

Author Response

Dear editors and reviewers. We would like to thank you for your effort and time spent in reviewing our manuscript. These are our individual responses to your comments:

Reviewer 1:

The manuscript entitled “Changes in dietary behaviours and lifestyle as risk factors for 2 weight gain during the COVID-19 lockdown in Chile: A cross-3 sectional study” presents interesting issue, however some corrections are needed

  • Introduction - In this section Authors presented the information associated with the COVID-19 pandemic. However, some detailed information about global situation is missing. This section should be briefly presented – what do we know and what is the background for this study. Some detailed information about other studies are necessary. The good background should present the history of problem, the current knowledge and scientific "gap", and then authors should present how their study could fill this gap to justify the study. The international context must be presented. Authors presented issues that may have influenced weight gain, but at the same time, there are numerous studies published that indicated the opposite situation associated with following a better diet during lock down, that should be also indicated, e.g. https://www.ncbi.nlm.nih.gov/pmc/articles/PMC7278251/; https://www.mdpi.com/2072-6643/12/9/2640.

R: Thank you for this suggestion. We have now addressed these issues in the introduction and included the mentioned study alongside others.

  • Line 77 - A sentence should never start with a number

R: We have now corrected this.

  • Materials and methods – how the recruitment was carried out?  How the subjects were selected? Why participants were recruited only from the university community?

R: Users of the institutional electronic mailing system, which included undergraduates, postgraduates, administrative personnel, and academics from the three campuses of University of Bío-Bío were invited to participate. These campuses are distributed across 2 regions in Chile: Ñuble and Bío-Bío.

Only participants from the university community were asked to respond the questionnaire since this work was part of the institutional study named: “Eating habits and lifestyles of students and employees from the university of Bío-Bío”. The main objective of this larger project is to identify early changes in lifestyle and wating habits that may affect the health of those who are part of universities form the Network of Universities Promoting Health.

We have now included in the title of our manuscript that this study includes participants from a university community from Chile to further clarify this.

  • Food Consumption Frequency Questionnaire; Emotional Eating Survey; Fantastic questionnaire; Household Food Insecurity Access Component Scale - Was the questionnaire previously validated? What was the accuracy and consistency of this questionnaire? What is the original language of the questionnaire? Was the questionnaire translated? Who did so? Any validation of the translated questionnaire?

R: All these questionnaires had been previously validated and applied in the Chilean population, as well as other populations. What we did was assemble these questionnaires into a single google forms document consisting of 85 items. Afterwards, we asked for this assembled questionnaire to be evaluated by as many experts as possible from the Department of  Nutrition and Public Health from the University of Bío-Bío. Thus, this assembled questionnaire was validated by experts in the field. Regarding individual validations of the questionnaires and their applications in Spanish and in the Chilean population:

  1. The Food Consumption Frequency Questionnaire was taken from the questionnaire applied in the National Food Consumption Study from the Ministry of Health of Chile which was conducted in the Chilean population in Spanish in the year 2010. References:
    • Herrera Figueroa Y. Norma general Técnica N° 148/2013 sobre Guías Alimentarias (GABA) para la Población Chilena, aprobada por Resolución N° 260/2013 del MINSAL. Santiago, Chile: Ministerio de salud; 2013.
    • Ministerio de salud, Gobierno de chile. Encuesta nacional de consumo alimentario. Chile: Facultad de medicina-Universidad de Chile Economía y Negocios-Universidad de Chile; 2014.p166-191. Disponible en: http://www.minsal.cl/sites/default/files/ANEXOS_ENCA.pdf
  2. The Emotional Eating Survey was originally developed in Spanish and applied in the Spanish population. This survey was posteriorly validated in the Chilean population. References:
    • Garaulet M, Canteras M, Morales E, López-Guimera G, Sánchez-Carracedo D, Corbalán-Tutau MD. Validación de un cuestionario sobre alimentación emocional para su uso en casos de obesidad: el Cuestionario Eater emocional (EEQ). Nutr Hosp . 2012; 27 (2): 645-651. doi: 10.1590 / S0212-16112012000200043
    • Gonzalez Mariela Validación del Cuestionario de Comedor Emocional (CCE) en Chile Revista GEN (Gastroenterología Nacional) 2018;72(1):21-24.
  3. The Fantastic Questionnaire was created by the Ministry of Health of Chile to measure lifestyles in the general population in an easy-to-answer manner. Thus, it was originally developed in Spanish and validated by both the Ministry of Health of Chile and other authors from Latin America. References:
    • ¿Cómo es tu estilo de vida? [Internet]. Buenas Prácticas APS. 2013 [cited 8 August 2020]. Available from: http://buenaspracticasaps.cl/wp-content/uploads/2013/10/Como_es_tu_estilo_de_vida.pdf
    • Betancurth Loaiza, Diana Paola, & Vélez Álvarez, Consuelo, & Jurado Vargas, Liliana (2015). Validación de contenido y adaptación del cuestionario Fantastico por técnica Delphi. Salud Uninorte, 31(2),214-227.
    • Ramírez-Vélez. R. & Agredo. R.A. (2012). Fiabilidad y validez del instrumento “Fantástico” para medir el estilo de vida en adultos colombianos. Revista Salud Pública, 14 (2), 226-237, 2012.
    • Barriga Silva, T. A. (2020). Instrumento "Fantástico" para medir el estilo de vida saludable de adolescentes de la comuna de Bulnes. Revista Reflexión E Investigación Educacional, 3(1), 61-74.
  4. The Household Food Insecurity Access Component Scale was originally developed in Spanish with cooperation form the Global Health Bureau of the United States Agency for International Development, to be applied in developing countries. This questionnaire has been extensively validated. References:
    • Coates, Jennifer, Anne Swindale y Paula Bilinsky. Escala del Componente de Acceso de la Inseguridad Alimentaria en el Hogar (HFIAS) para la Medición del Acceso a los Alimentos en el Hogar: Guía de Indicadores (v. 2). Washington, D.C.: Proyecto de Asistencia Técnica sobre Alimentos y Nutrición, Academia para el Desarrollo Educativo, agosto de 2007
    • Salvador G, Ngo J, Pérez C, Aranceta J. Escalas de evaluación de la inseguridad alimentaria en el hogar. Rev Esp Nutr Comunitaria. 2015;21: 270-276.
    • Ballard, T., Kepple, A., & Cafiero, C. (2013). The Food Insecurity Experience Scale. Development of a Global Standard for Monitoring Hunger Worldwide. (FAO, Ed.) Technical Paper. Rome, FAO.
    • Cafiero C, Melgar-Quiñonez HR, Ballard TJ, Kepple AW. Validity and reliability of food security measures. Ann N Y Acad Sci. 2014 Dec; 1331:230-248. doi: 10.1111/nyas.12594.
    • Castell, G., De la Cruz, J., Pérez, C., & Aranceta, J. (2015). Escalas de evaluación de la inseguridad alimentaria en el hogar. Revista Española de Nutrición, 21(1), 270-276.
    • Féliz-Varduzco, G., Aboites-Manrique, G., & Castro-Lugo, D. (2018). La seguridad alimentaria y su relación con la suficiencia e incertidumbre del ingreso: un análisis de las percepciones del hogar. Acta Universitaria, 28(4), 74-86

  • Lines 109-110 - Was the normality of distribution tested? The information about it should be added and authors should be consequent. If data have normal distribution, they should be treated as such, if not, nonparametric tests should be applied. Please specify it.

R: We have now added this clarification in the manuscript. The distribution of the sample was evaluated through kurtosis and asymmetry and concluded to have a normal distribution with values for kurtosis ±2 and asymmetry ±0.5.

  • Table 1 – please remove information about weight (please remain only the data of BMI). It is misleading – it is natural that the weight of men and women differ!

R: Thank you for this suggestion, we eliminated this form the table.  

  • Tables are a little bit difficult to follow – please improve it.  

R: The current design of the tables has been worked out by the editorial for this revised version of the manuscript. We have further made some style changes which could aid visualization of the tables.

  • Please unify the number of decimal places in p-Values

R: We have corrected this accordingly.

  • Authors should in their discussion include 3 areas: (1) compare gathered data with the results by other authors, (2) formulate implications of the results of their study and studies by other authors, (3) formulate the future areas which should be studied.
  • Authors should present here and discuss the limitations of their study.
  • In limitation section (at the end of the discussion section) authors should add the information about low share of men in the sample (as a bias)

R: We have included all these points in the discussion.

Reviewer 2 Report

Manuscript title: Changes in dietary behaviours and lifestyle as risk factors for weight gain during the COVID-19 lockdown in Chile: A cross-sectional study

The study was cross-sectional. The researchers collected data on 639 individuals. The authors state that the purpose was to evaluate the effect of changes in dietary behaviours and lifestyle on weight gain during the COVID-19 lockdown in Chile. The investigation included several interesting findings, but there were also some significant problems.

TITLE: The title includes some words that are not accurate. First, “change” requires the passage of time. This was a cross-sectional study, so it was not prospective. Perhaps “changes” should be replaced with “differences”? Additionally, risk is synonymous with incidence. Technically, a variable cannot be a risk factor unless it is measured prospectively. The variables investigated in this study were correlates, but not risk factors. It was good that the authors included mention that the study was cross-sectional, but this fact is not consistent with the other issues mentioned above.

ABSTRACT: The authors state, “We aimed to evaluate the effect of changes in dietary behaviours and lifestyle on weight gain during the COVID-19 lockdown in Chile.” “Effect” is misused in this statement. Effect implies causation, like cause-and-effect. Since this was a correlational study, the authors could not evaluate “effect.”

Additionally, the authors state that mean weight gain was 1.99 kg… Readers will likely interpret this as measured weight gain, whereas it was self-reported. It should be reported as “self-reported” throughout the paper. Also, means should not be reported without a measure of variation, like standard deviation.

The authors indicate “The increases in dietary intake…” For there to be an “increase,” a study must be prospective. Therefore, the statement(s) should be reworded so that “increase” is replaced with something like “pre-COVID to COVID differences in food groups…”

The word “increase” is used not only in the Abstract, but throughout the paper. Again, there cannot be an increase without a prospective design.

The authors state, “the main risk factors associated with weight gain.” Again, the researchers did not evaluate risk, so these variables were not risk factors. Correlation is not the same as risk.

INTRODUCTION:

The authors use semi-colons (;) throughout the paper. In virtually all of these cases, ; should be replaced with a period (.) and a new sentence should be started after the period.

The authors state, “Understanding how lockdowns modify lifestyle, feeding behavior, and emotional influence on dietary habits…” This sentence immediately precedes their statement of the problem. However, this statement uses the word “modify,” which is a causal verb. It is used to mean “influence,” yet the present study does not and cannot address influence or cause.

Line 59. Again, “change” requires time to pass. This was a cross-sectional study. Also, a key word is missing from this sentence. “Changes do not happen… “on weight gain.” This statement of the purpose needs to use words denoting correlation or relationships, not words implying risk or cause.

The Introduction is short and does not build a meaningful case or justification for the study.
Additional background about isolation and confinement and their relationships with dietary behaviors and lifestyle would be helpful. The authors discuss lockdowns a bit in the Discussion. Perhaps some of these concepts can be worked into the Introduction?

MATERIALS AND METHODS

Quality research requires that key variables be reliable and valid. The authors do not provide evidence that the variables used in this study were reliable and valid.

Line 70. How was the survey validated by experts in nutrition and public health? What type of validation research was conducted? Were the surveys instruments found to be reliable? Was there concurrent or predictive validity? Has the survey been used and published in other research?

Line 71. A sample was used, but not a sample population. Samples and populations are two different groups.

Why was the sample confined to a university? Are universities representative of the general population?

Line 77. Faculty, administrators, and students are very different individuals. How were the 1000 individuals selected? Why was a sample of 1000 chosen? Was an equal number from each subgroup invited to participate?

Line 80. …emotional influence on feeding behavior,… could not be measured. To measure “influence” requires a randomized controlled trial.

Line 81.  To measure “increases” requires at least two time periods. This study was cross-sectional, so there was no passage of time. Perhaps “differences” were measured? This problem occurs throughout the paper.

Lines 92-94. It is not clear how the CFCA instrument was used to measure eating habits before and during the lockdown.

Scores on the Emotional Eating Survey were categorized. Why weren’t continuous values used?  Have these categories been used before? Similarly, categories were used to score the Fantastic Questionnaire. Why?

Line 105. Is the “19” following the citation supposed to be a citation as well or is this a typo?

Line 115. A decrease in food intake could not be measured with a cross-sectional design.

Lines 112-116. It appears that mean differences were determined without controlling for differences in age, sex, job (faculty, administration, or student, etc), etc. It was good that age and sex were controlled later (Lines 137-138). Differences between students and faculty would seem to be a critical factor and need to be controlled throughout?

Lines 125 and 130 and 135. Risk cannot be determined unless a prospective design is used.

Line 143- the impact of lifestyle changes could not be determined in this study. Impact is a word meaning influence or cause and change requires a prospective design.

Line 147-148. The authors mention use of the Bonferroni test. It appears to be used to adjust only for a couple of statistical tests associated with emotional well-being. Specifically, what values were divided by what numbers? Moreover, numerous statistical tests were conducted in this study. What was done to control for the inflation of Type I error given the extreme number of tests that were administered? Bonferroni is an ultra-conservative approach, but other methods are available to reduce the likelihood of committing a Type I error and should be employed.

What percent of students, faculty, and administrators were asked to answer the questionnaire? What percent of each group participated? Was this mostly a survey of university students?

As mentioned many times previously, changes and increases/decreases were not measured. This was not a prospective study.

In Table 1, what does Datos sociodemográficos y de hábitos alimentarios mean?

Numerous variables are included, but they have not been defined or explained.

Line 183. “whit” appears to be a typo.

Line 195. Just because two variables are correlated does not make one a risk factor of the other.

DISCUSSION

Again, it is important that differences in body weight and BMI from before and during the lockdown be labeled as self-reported.

Lines 233-234. It is good that the authors mention a few previous studies about weight gain. More evidence and support using previous studies associated with specific lifestyle factors would be appropriate in the Discussion.

Lines 240-241. The authors state, “Furthermore, we found that as the frequency of consumption of industrialized foods (cookies and sweet or filled cakes) increases, the risk of weight gain also increased. This important statement and many others in the paper suggest a prospective design, even though the study was cross-sectional.

Line 247. The authors state, “…although the emotional influence seemingly affected BMI…” This study was not designed to determine if emotional well-being affected BMI. “Affected” means influenced.

Another major limitation of the study was that numerous statistical tests were administered, increasing the probability that relationships and differences were identified as statistically significant, but were due to chance.

The authors do a good job of focusing on the concept of associations rather than risk or influence in their conclusion. However, throughout the conclusion they use the term “change” which gives the impression that the study was prospective. Terminology such as pre-COVID to COVID differences would help to minimize the tendency to make the results prospective rather than cross-sectional.

Thank you for your many hours of work in developing this study!

Author Response

Dear editors and reviewers. We would like to thank you for your effort and time spent in reviewing our manuscript. These are our individual responses to your comments:

Reviewer 2:

Manuscript title: Changes in dietary behaviours and lifestyle as risk factors for weight gain during the COVID-19 lockdown in Chile: A cross-sectional study

The study was cross-sectional. The researchers collected data on 639 individuals. The authors state that the purpose was to evaluate the effect of changes in dietary behaviours and lifestyle on weight gain during the COVID-19 lockdown in Chile. The investigation included several interesting findings, but there were also some significant problems.

TITLE: The title includes some words that are not accurate. First, “change” requires the passage of time. This was a cross-sectional study, so it was not prospective. Perhaps “changes” should be replaced with “differences”? Additionally, risk is synonymous with incidence. Technically, a variable cannot be a risk factor unless it is measured prospectively. The variables investigated in this study were correlates, but not risk factors. It was good that the authors included mention that the study was cross-sectional, but this fact is not consistent with the other issues mentioned above.

R: Thank you for the detailed explanation and suggestion. We have now corrected the title accordingly as well as all other sections in the manuscript which could lead to this misinterpretation. We have replaced the words “changes” for “differences”, and “risk” for “associated factors” or “odds”. Since we did not perform correlation tests, but we applied statistical methods to test associations, we chose to use the term “association” rather than “correlation” as suggested.

ABSTRACT: The authors state, “We aimed to evaluate the effect of changes in dietary behaviours and lifestyle on weight gain during the COVID-19 lockdown in Chile.” “Effect” is misused in this statement. Effect implies causation, like cause-and-effect. Since this was a correlational study, the authors could not evaluate “effect.”

R: We substituted the term “effect” for “association”.

Additionally, the authors state that mean weight gain was 1.99 kg… Readers will likely interpret this as measured weight gain, whereas it was self-reported. It should be reported as “self-reported” throughout the paper. Also, means should not be reported without a measure of variation, like standard deviation.

R: We have now corrected this as suggested.

The authors indicate “The increases in dietary intake…” For there to be an “increase,” a study must be prospective. Therefore, the statement(s) should be reworded so that “increase” is replaced with something like “pre-COVID to COVID differences in food groups…”

R: We decided to substitute “increase in dietary intake” for “greater self-reported dietary intake”. We decided not to include pre-COVID-19 and post-COVID-19 since readers could incorrectly interpret these terms as before having clinical COVID-19 and after having been diagnosed with COVID-19. In cases for which the suggested clarification was consider important to avoid misinterpretation, we added pre-lockdown and post-lockdown to further clarify.

The word “increase” is used not only in the Abstract, but throughout the paper. Again, there cannot be an increase without a prospective design.

R: We have made the appropriate corrections to avoid this misinterpretation.

The authors state, “the main risk factors associated with weight gain.” Again, the researchers did not evaluate risk, so these variables were not risk factors. Correlation is not the same as risk.

R: We have avoided using the work “risk” and we have included the term “association” as we have already clarified for one of the previous comments.

INTRODUCTION:

The authors use semi-colons (;) throughout the paper. In virtually all of these cases, ; should be replaced with a period (.) and a new sentence should be started after the period.

R: We have substituted semi-colons for periods when appropriate.

The authors state, “Understanding how lockdowns modify lifestyle, feeding behavior, and emotional influence on dietary habits…” This sentence immediately precedes their statement of the problem. However, this statement uses the word “modify,” which is a causal verb. It is used to mean “influence,” yet the present study does not and cannot address influence or cause.

R=We modified this phrase to avoid this problem.

Line 59. Again, “change” requires time to pass. This was a cross-sectional study. Also, a key word is missing from this sentence. “Changes do not happen… “on weight gain.” This statement of the purpose needs to use words denoting correlation or relationships, not words implying risk or cause.

R: Thank you for observing this. We have made the appropriate modifications to avoid incorrectly implying of suggesting causation.

The Introduction is short and does not build a meaningful case or justification for the study.
Additional background about isolation and confinement and their relationships with dietary behaviors and lifestyle would be helpful. The authors discuss lockdowns a bit in the Discussion. Perhaps some of these concepts can be worked into the Introduction?

R: Thank you, we have now added more significant background of existing research on this topic.

MATERIALS AND METHODS

Quality research requires that key variables be reliable and valid. The authors do not provide evidence that the variables used in this study were reliable and valid. Line 70. How was the survey validated by experts in nutrition and public health? What type of validation research was conducted? Were the surveys instruments found to be reliable? Was there concurrent or predictive validity? Has the survey been used and published in other research?

R: This issue was also highlighted by reviewer 1. We have already responded to this with the following answers:

Users of the institutional electronic mailing system, which included undergraduates, postgraduates, administrative personnel, and academics from the three campuses of University of Bío-Bío were invited to participate. These campuses are distributed across 2 regions in Chile: Ñuble and Bío-Bío.

Only participants from the university community were asked to respond the questionnaire since this work was part of the institutional study named: “Eating habits and lifestyles of students and employees from the university of Bío-Bío”. The main objective of this larger project is to identify early changes in lifestyle and wating habits that may affect the health of those who are part of universities form the Network of Universities Promoting Health.

We have now included in the title of our manuscript that this study includes participants from a university community from Chile to further clarify this.

  • Food Consumption Frequency Questionnaire; Emotional Eating Survey; Fantastic questionnaire; Household Food Insecurity Access Component Scale - Was the questionnaire previously validated? What was the accuracy and consistency of this questionnaire? What is the original language of the questionnaire? Was the questionnaire translated? Who did so? Any validation of the translated questionnaire?

R: All these questionnaires had been previously validated and applied in the Chilean population, as well as other populations. What we did was assemble these questionnaires into a single google forms document consisting of 85 items. Afterwards, we asked for this assembled questionnaire to be evaluated by as many experts as possible from the Department of  Nutrition and Public Health from the University of Bío-Bío. Thus, this assembled questionnaire was validated by experts in the field. Regarding individual validations of the questionnaires and their applications in Spanish and in the Chilean population:

  1. The Food Consumption Frequency Questionnaire was taken from the questionnaire applied in the National Food Consumption Study from the Ministry of Health of Chile which was conducted in the Chilean population in Spanish in the year 2010. References:
    • Herrera Figueroa Y. Norma general Técnica N° 148/2013 sobre Guías Alimentarias (GABA) para la Población Chilena, aprobada por Resolución N° 260/2013 del MINSAL. Santiago, Chile: Ministerio de salud; 2013.
    • Ministerio de salud, Gobierno de chile. Encuesta nacional de consumo alimentario. Chile: Facultad de medicina-Universidad de Chile Economía y Negocios-Universidad de Chile; 2014.p166-191. Disponible en: http://www.minsal.cl/sites/default/files/ANEXOS_ENCA.pdf
  2. The Emotional Eating Survey was originally developed in Spanish and applied in the Spanish population. This survey was posteriorly validated in the Chilean population. References:
    • Garaulet M, Canteras M, Morales E, López-Guimera G, Sánchez-Carracedo D, Corbalán-Tutau MD. Validación de un cuestionario sobre alimentación emocional para su uso en casos de obesidad: el Cuestionario Eater emocional (EEQ). Nutr Hosp . 2012; 27 (2): 645-651. doi: 10.1590 / S0212-16112012000200043
    • Gonzalez Mariela Validación del Cuestionario de Comedor Emocional (CCE) en Chile Revista GEN (Gastroenterología Nacional) 2018;72(1):21-24.
  3. The Fantastic Questionnaire was created by the Ministry of Health of Chile to measure lifestyles in the general population in an easy-to-answer manner. Thus, it was originally developed in Spanish and validated by both the Ministry of Health of Chile and other authors from Latin America. References:
    • ¿Cómo es tu estilo de vida? [Internet]. Buenas Prácticas APS. 2013 [cited 8 August 2020]. Available from: http://buenaspracticasaps.cl/wp-content/uploads/2013/10/Como_es_tu_estilo_de_vida.pdf
    • Betancurth Loaiza, Diana Paola, & Vélez Álvarez, Consuelo, & Jurado Vargas, Liliana (2015). Validación de contenido y adaptación del cuestionario Fantastico por técnica Delphi. Salud Uninorte, 31(2),214-227.
    • Ramírez-Vélez. R. & Agredo. R.A. (2012). Fiabilidad y validez del instrumento “Fantástico” para medir el estilo de vida en adultos colombianos. Revista Salud Pública, 14 (2), 226-237, 2012.
    • Barriga Silva, T. A. (2020). Instrumento "Fantástico" para medir el estilo de vida saludable de adolescentes de la comuna de Bulnes. Revista Reflexión E Investigación Educacional, 3(1), 61-74.
  4. The Household Food Insecurity Access Component Scale was originally developed in Spanish with cooperation form the Global Health Bureau of the United States Agency for International Development, to be applied in developing countries. This questionnaire has been extensively validated. References:
    • Coates, Jennifer, Anne Swindale y Paula Bilinsky. Escala del Componente de Acceso de la Inseguridad Alimentaria en el Hogar (HFIAS) para la Medición del Acceso a los Alimentos en el Hogar: Guía de Indicadores (v. 2). Washington, D.C.: Proyecto de Asistencia Técnica sobre Alimentos y Nutrición, Academia para el Desarrollo Educativo, agosto de 2007
    • Salvador G, Ngo J, Pérez C, Aranceta J. Escalas de evaluación de la inseguridad alimentaria en el hogar. Rev Esp Nutr Comunitaria. 2015;21: 270-276.
    • Ballard, T., Kepple, A., & Cafiero, C. (2013). The Food Insecurity Experience Scale. Development of a Global Standard for Monitoring Hunger Worldwide. (FAO, Ed.) Technical Paper. Rome, FAO.
    • Cafiero C, Melgar-Quiñonez HR, Ballard TJ, Kepple AW. Validity and reliability of food security measures. Ann N Y Acad Sci. 2014 Dec; 1331:230-248. doi: 10.1111/nyas.12594.
    • Castell, G., De la Cruz, J., Pérez, C., & Aranceta, J. (2015). Escalas de evaluación de la inseguridad alimentaria en el hogar. Revista Española de Nutrición, 21(1), 270-276.
    • Féliz-Varduzco, G., Aboites-Manrique, G., & Castro-Lugo, D. (2018). La seguridad alimentaria y su relación con la suficiencia e incertidumbre del ingreso: un análisis de las percepciones del hogar. Acta Universitaria, 28(4), 74-86

Line 71. A sample was used, but not a sample population. Samples and populations are two different groups.

R: Thank you for this clarification. We have now corrected it.

Why was the sample confined to a university? Are universities representative of the general population?

R: It would be incorrect to assume that this university community is representative of the general population. Therefore, we have now modified the title and abstract to further clarify that this study was performed in a university community and may not be representative of the general population.

Line 77. Faculty, administrators, and students are very different individuals. How were the 1000 individuals selected? Why was a sample of 1000 chosen? Was an equal number from each subgroup invited to participate?

R: This statement is true since the type of participants are different. However as previously explained, it was important to include all the University Community due to the main objective of the University Project which is to procure health of all people who belong to the University Community. The total number of individuals from the University Community is approximately 3300, composed by ~2000 students and ~1300 employees including academics and administrative officials (this is an approximate since the numbers tend to vary throughout the year since the University Community is dynamic). Of these, only approximately 1000 are considered active users of the electronic mailing system, which is why the sample for this study was composed of 1000 potential participants who were sent the invitation to respond the questionnaire.

Line 80. …emotional influence on feeding behavior,… could not be measured. To measure “influence” requires a randomized controlled trial.

R: We have corrected this phrase as well.

Line 81.  To measure “increases” requires at least two time periods. This study was cross-sectional, so there was no passage of time. Perhaps “differences” were measured? This problem occurs throughout the paper.

R: We have corrected this throughout the manuscript.

Lines 92-94. It is not clear how the CFCA instrument was used to measure eating habits before and during the lockdown.

R=We have now further clarified in the manuscript how this instrument was used.  

Scores on the Emotional Eating Survey were categorized. Why weren’t continuous values used?  Have these categories been used before? Similarly, categories were used to score the Fantastic Questionnaire. Why?

R: The original questionnaire was developed by using qualitative variables; therefore, it does not quantify emotional status. There are 4 possible answers for every item: 1) never, 2) sometimes, 3) generally, and 4) always. For every answer points are added. With lower total scores, the behaviour is healthier. In the original instrument, total scores are categorised into the used categories. This is why we decided to be compliant with the original survey. Similarly, we used the original categories described for the fantastic questionnaire.

Line 105. Is the “19” following the citation supposed to be a citation as well or is this a typo?

  1. This was a typo, we apologize for it.

Line 115. A decrease in food intake could not be measured with a cross-sectional design.

R: We have corrected this recurrent problem throughout the manuscript.  

Lines 112-116. It appears that mean differences were determined without controlling for differences in age, sex, job (faculty, administration, or student, etc), etc. It was good that age and sex were controlled later (Lines 137-138). Differences between students and faculty would seem to be a critical factor and need to be controlled throughout?

R: Thank you for this suggestion. We have now included occupation for adjustment of all regression models. After re-analyses, the new results were included in substitution of the prior ones. Frequencies and percentages for occupation were also included in table 1.

Lines 125 and 130 and 135. Risk cannot be determined unless a prospective design is used.

R: We substituted the word “risk” for “odds”.

Line 143- the impact of lifestyle changes could not be determined in this study. Impact is a word meaning influence or cause and change requires a prospective design.

R: We have corrected this by avoiding the word “impact”.

Line 147-148. The authors mention use of the Bonferroni test. It appears to be used to adjust only for a couple of statistical tests associated with emotional well-being. Specifically, what values were divided by what numbers? Moreover, numerous statistical tests were conducted in this study. What was done to control for the inflation of Type I error given the extreme number of tests that were administered? Bonferroni is an ultra-conservative approach, but other methods are available to reduce the likelihood of committing a Type I error and should be employed.

R: The Bonferroni test we applied was a post hoc analysis after the random effects model. This test was chosen since it allows to counter the potential error arisen from multiple comparisons by adjusting the confidence interval for every individual comparison depending on the number of comparisons performed. This post hoc test is recommended for random effects models due to its conservativeness and since it works well for samples with homoscedasticity and few multiple comparisons. We added a brief clarification in the manuscript to specify that this is a post hoc analysis for this specific model and not a Bonferroni adjustment for all other statistical analysis. Test or adjustment was not necessary for al other analyses since we did not make multiple comparisons between groups for all quantitative comparisons.

We were careful to avoid type 1 error by avoiding creating subgroups for the sample and by avoiding as much as possible performing excessive comparisons. The statistical analyses were applied according to the methodological design and statistical plan. For comparisons between men and women, comparisons were performed among the same subgroup of subjects. For individuals with similar food intake and higher food intake, comparisons were only performed between both groups considering the total sample and not by subgroups. Only the random effects model required adjustment of p values according to the number of comparisons, for which we used the post Hoc Bonferroni test as explained earlier. We considered the variables with the greater effect size for the previous models, to avoid erroneously finding differences due to extensive reanalysis. All regression models were performed through methods of maximal verisimilitude, Logistic regression models were applied trough univariable and multivariable models, through the Stepwise method, which systematically chooses the better model with the introduction of every variable with an ANOVA model, which allows to have certainty that the best final model is chosen. Lastly, the multinomial model allows to identify the associations for every variable not collapsed into a single category and p values are directly adjusted for the model when two consecutive comparisons are performed.

What percent of students, faculty, and administrators were asked to answer the questionnaire? What percent of each group participated? Was this mostly a survey of university students?

R: We have now included in the table of descriptive results the occupation of participants of this study.

As mentioned many times previously, changes and increases/decreases were not measured. This was not a prospective study.

R: We corrected this throughout the manuscript.

In Table 1, what does Datos sociodemográficos y de hábitos alimentarios mean?

R= We are sorry for this phrase which was left untranslated. We have now corrected it.

Numerous variables are included, but they have not been defined or explained.

R: We added a paragraph in the material and methods section to define independent and dependent variables.

Line 183. “whit” appears to be a typo.

R=Thank you, we corrected this typo.

Line 195. Just because two variables are correlated does not make one a risk factor of the other.

R: We have avoided the word “risk factor”.

DISCUSSION

Again, it is important that differences in body weight and BMI from before and during the lockdown be labeled as self-reported.

R: We have labelled this as suggested throughout the manuscript.

Lines 233-234. It is good that the authors mention a few previous studies about weight gain. More evidence and support using previous studies associated with specific lifestyle factors would be appropriate in the Discussion.

R: We added a sentence referring to specific lifestyle factors which have been associated with weight gain during lockdown.

Lines 240-241. The authors state, “Furthermore, we found that as the frequency of consumption of industrialized foods (cookies and sweet or filled cakes) increases, the risk of weight gain also increased. This important statement and many others in the paper suggest a prospective design, even though the study was cross-sectional.

R: We have corrected this in the manuscript.

Line 247. The authors state, “…although the emotional influence seemingly affected BMI…” This study was not designed to determine if emotional well-being affected BMI. “Affected” means influenced.

R: We have corrected this by substituting it for association.

Another major limitation of the study was that numerous statistical tests were administered, increasing the probability that relationships and differences were identified as statistically significant, but were due to chance.

R: Thank you for this suggestion. We have previously commented on the need and appropriateness for all the statistical analyses performed. However, we agree that it is important to advise readers to interpret differences with reserve since the probability that some of the findings are due to chance is always patent. We have now added this limitation in the discussion of our study.

The authors do a good job of focusing on the concept of associations rather than risk or influence in their conclusion. However, throughout the conclusion they use the term “change” which gives the impression that the study was prospective. Terminology such as pre-COVID to COVID differences would help to minimize the tendency to make the results prospective rather than cross-sectional.

R=Thank you for this suggestion. We have made these clarifications by using the terms “self-reported”, “pre-lockdown”, and “during-lockdown” as priorly described.  

Thank you for your many hours of work in developing this study!

Round 2

Reviewer 1 Report

The manuscript entitled “ Association of differences in dietary behaviours and 2lifestyle with self-reported weight gain during the COVID-19 3lockdown in a university community from Chile: A cross-sectional study” (new title) presents interesting issue. I appreciate the great efforts that the authors have made in response to my questions and concerns. However, there are some issues that should be corrected:

  • Table 1 – please remove information about height (similar as you did in case of weight). It is natural that the weight of men and women differ!
  • Figure 1 should be improved (in the presented form it is difficult to read it and interpret)

Author Response

Dear editors and reviewers. We would like to thank you once more for your effort and time spent in reviewing our manuscript. These are our individual responses to your comments:

The manuscript entitled “ Association of differences in dietary behaviours and lifestyle with self-reported weight gain during the COVID-19 lockdown in a university community from Chile: A cross-sectional study” (new title) presents interesting issue. I appreciate the great efforts that the authors have made in response to my questions and concerns. However, there are some issues that should be corrected:

Table 1 – please remove information about height (similar as you did in case of weight). It is natural that the weight of men and women differ!

R: Thank you for suggesting this. We agree with this observation, and we have now removed height from the table.

Figure 1 should be improved (in the presented form it is difficult to read it and interpret)

R: We have now improved visualization of Figure 1 by: 1. Changing the graphical direction of visual elements which are now in vertical order (vs prior horizontal order), and 2. We increased text size for labels in this figure.